# WORD EMBEDDING RE-EXAMINED: THEORETICAL ANALYSIS ABOUT WORD SIMILARITY AND ANALOGY STRUCTURE

## ABSTRACT

As observed in previous works, many word embedding methods exhibit two interesting properties: (1) words having similar semantic meanings are embedded closely; (2) analogy structure exists in the embedding space, such that "*Paris* is to *France* as *Berlin* is to *Germany*". We theoretically analyze the inner mechanism leading to these nice properties. Specifically, the embedding can be viewed as a low rank transformation from the word-context co-occurrence space to the embedding space. Such embedding transformation will preserve the relative distances among words. Furthermore, previous studies empirically observed that the parameter $\alpha$ has a strong influence on the performance of word embedding but did not provide theoretical explanation. We provide a theoretical explanation for this behavior, and derive a method to automatically find its optimal value. The experiments on real datasets verify our analysis.

## 1 INTRODUCTION

Word embedding is a fundamental task in natural language processing. Many different approaches have been proposed, including LSA (Deerwester et al., 1990), SGNS (Mikolov et al., 2013a;b), GloVe (Pennington et al., 2014) and others. These methods have achieved huge success in information retrieval (Salton & Buckley, 1988), entity recognition (Lample et al., 2016), sentiment analysis (Socher et al., 2013), machine translation (Sutskever et al., 2014) and so on.

Previous studies have demonstrated three well-known facts about the word embedding: (1) Levy & Goldberg (2014b) showes many different kinds of word embeddings (e.g. SGNS) can be understood as the matrix factorization framework. For example, Levy & Goldberg (2014b) proves that SGNS is implicitly factorizing the shifted pointwise mutual information (PMI) matrix. (2) Mikolov et al. (2013b); Pennington et al. (2014); Levy et al. (2015); Tian et al. (2016) and other works observe that word embedding exhibits two nice properties: the word similarity such that words with similar semantic meanings are embedded closely in the embedding space, and the analogy structure such that "woman is to queen as man is to king". (3) Previous studies, including (Caron, 2001; Turney, 2012; Bullinaria & Levy, 2012; Levy et al., 2015; Artetxe et al., 2018), *empirically* observe that $\alpha$ in word embedding $\boldsymbol{E} = \boldsymbol{U}_d \cdot \boldsymbol{\Sigma}_d^\alpha$ [1] has an important influence on the quality of word embedding. But these works did not provide theoretical explanation for this behavior of $\alpha$ or practical guide to set this parameter.

In this paper, we show the word embedding can be understood as a low rank transformation process. Furthermore, we investigate the change of the relative distances between words which is the inner mechanism that leads to these two nice properties in the word embedding. Particularly, we find the parameter $\alpha$ has influence on the change of relative distance between words. Based on such analysis, we provide theoretical explanation the behavior of $\alpha$ and derive a method to automatically find its optimal value. To summarize, this research is motivated by two questions:

1. What is the inner mechanism resulting in these two good properties of word embedding?

2. How does the parameter $\alpha$ influence the word embedding? Can we find a method to determine its optimal value?

---

[1] We will explain the meaning of these variables (e.g., $\boldsymbol{U}_d$) in the following sections.

We make the following contributions in this paper:

- We reveal how the relative distances between words change during the embedding process which can be seen as a low rank transformation process. The nice properties of word embeddings are inherited from the original word-context co-occurrence matrix as the result of such low rank transformation.

- Compared with previous studies which *empirically* observed the influence of the parameter $\alpha$ on the word embedding, we theoretically explain the influence of parameter $\alpha$ on the word embedding by revealing it controls the change of relative distances between words. Besides, we derive a method to automatically find its optimal value. We conduct experiments on real datasets to verify our ideas.

The following sections are organized as follows: section 2 introduces the background knowledge and necessary preliminaries; section 3 shows how the relative distances between words change during this linear transformation and it is the inner reason incurring these nice properties of word embedding. Based on such analysis, section 4 shows that symmetric factorization can be suboptimal and how to improve the word embedding model with asymmetric factorization. Section 5 contains the experiments on real datasets and section 6 discusses the related work. Finally, section **??** is the conclusion and describes a potential work.

## 2 BACKGROUND AND PRELIMINARIES

The goal of word embedding is to find a good vector representation for every word in a corpus. As summarized in Yin & Shen (2018), most existing word embedding methods can be formulated as low rank matrix approximations. So we can analyze the word embedding algorithms (e.g., SGNS) within the matrix factorization framework. Suppose $M \in \mathbb{R}^{n \times n}$ is the word-context matrix where $M_{i,j}$ represents some statistics between word $w_i$ and its context word $w_j$. These word embedding algorithms are seeking a $d$-dimension word embedding matrix $E \in \mathbb{R}^{n \times d}$ accompanied with a context embedding matrix $C \in \mathbb{R}^{n \times d}$ to approximate the word-context matrix $M$, such that $M \approx EC^T$. The matrix $M$ can have different forms due to the design of objective function in every embedding method. For example, $M$ is the shifted pointwise mutual information matrix in the SGNS.

**Explicitly performing singular value decomposition.** This kind of methods can be summarized as follows. Firstly, a word-context matrix $M$ is constructed to capture some co-occurrence statistics of every word and its context. For example, $M$ can be the pointwise mutual information (PMI) matrix (Church & Hanks, 1990), positive PMI (PPMI) matrix (Bullinaria & Levy, 2012) and Shifted PPMI (SPPMI) matrix (Levy & Goldberg, 2014b). Then, SVD is used to factorize the word-context matrix $M$ as $M = U \cdot \Sigma \cdot V^T$. Finally, only the top $d$ singular values in $\Sigma$ and the corresponding columns in $U$ and $V$ are kept, producing $\Sigma_d$, $U_d$ and $V_d$ respectively. The word embedding $E \in \mathbb{R}^{n \times d}$ is obtained by $E = U_d$ or $E = U_d \cdot \Sigma_d$.
Caron (2001), Turney (2012), Bullinaria & Levy (2012), Levy et al. (2015) and Artetxe et al. (2018) discussed a more general approach which adds a parameter $\alpha$ to the truncated diagonal matrix $\Sigma_d$ and obtain the word embedding $E = U_d \cdot \Sigma_d^\alpha$. They observe $\alpha$ has an important influence on the embedding and suggest this parameter should be tuned. However, the reason is not theoretically clear and no clear method has been provided to tune $\alpha$ yet.

**Implicitly performing matrix factorization.** Recently, word embedding methods based on neural networks have been proposed (e.g., Bengio et al. (2003)). Particularly, Mikolov et al. (2013a;b) proposed a popular word embedding approach SGNS based on the skip-gram with negative-sampling and it achieves the state-of-the-art results in different tasks. Besides, GloVe is another widely used word embedding method (Pennington et al., 2014). While SGNS and GloVe learn word embeddings by optimizing some objective functions using stochastic gradient methods, it has been shown that these two methods are implicitly performing matrix factorizations. Specifically, Levy & Goldberg (2014b) showed SGNS is implicitly factorizing the shifted Pointwise Mutual Information matrix. Levy et al. (2015) showed the GloVe is implicitly factorizing the log-count word-context co-occurrence matrix. So these two neural network based embedding methods can also be formu-

lated within the matrix factorization framework, and the parameter $\alpha$ equals to $0.5$ as they are doing symmetric factorization of some word-context co-occurrence matrix $M$.

To summarize, same as the first assumption in Yin & Shen (2018), our analysis assumes the word embeddings can be formulated as low rank matrix approximations, either explicitly or implicitly. Specifically, the embedding matrix $E$ equals to $U_d \cdot \Sigma_d^\alpha$ where $U_d$, $\Sigma_d$ and $\alpha$ are as defined before.

# 3 ANALYSIS: THE REASON WHY THE EMBEDDING EXHIBITS NICE PROPERTIES

Our start point is the relation between word embedding and matrix factorization. As mentioned in section 2, almost all word embedding methods can be formulated as explicit or implicit matrix factorization (e.g., SGNS is implicitly doing a symmetric factorization on the shifted pointwise mutual information matrix). We adopt the same notation in previous papers (Levy et al., 2015; Yin & Shen, 2018) to write $M = U\Sigma V^T$ as the SVD of some word-context co-occurrence matrix $M$. For example, $M$ can be the pointwise mutual information matrix. Besides, $\Sigma_d$ is the truncated diagonal matrix containing top $d$ singular values. $U_d$ and $V_d$ are the corresponding truncated $U$ and truncated $V$ respectively. The $d$-dimension word embedding matrix $E$ is obtained by multiplying $U_d$ with a power of $\Sigma_d$,

$$E = U_d \cdot \Sigma_d^\alpha \tag{1}$$

## 3.1 WORD EMBEDDING AS LOW RANK TRANSFORMATION

The embedding $E$ in equation 1 is indeed a low rank transformation from the original word-context co-occurrence space $M$. Let us define another diagonal matrix $\Sigma_{pse} = \text{diag}(\sigma_1, \ldots, \sigma_d, 0, \ldots, 0)$ whose first $d$ elements are the singular values in $\Sigma$ and the remaining diagonal elements are zeros. We have the following observation.

**Observation 1.** $E$ is the first $d$ columns of $E_{pse}$

$$E_{pse} = M \cdot V \cdot \Sigma_{pse}^{(\alpha-1)} \tag{2}$$

*The proof is in Appendix A.1.*

Observation 1 reveals the process to compute $E$: (1) first multiplying the matrix $V$ to the right side of $M$; (2) then multiplying the diagonal matrix $\Sigma_{pse}^{(1-\alpha)}$ on the intermediate result $MV$; (3) finally removing the last $n - d$ columns of $M \cdot V \cdot \Sigma_{pse}^{(\alpha-1)}$. The first step in this transformation process is a rotation because the matrix $V$ is a unitary matrix. So the relative distances between words are not changed during the first transformation. In the second step, the matrix $\Sigma_{pse}$ is a diagonal matrix and it is multiplied on the right side of $MV$. So the second step scales every dimension of the rotated matrix. Because we keep the first $d$ dimensions in the final step, we only care about the scaling in the first $d$ dimensions which are determined by the corresponding $d$ singular values with a power $\alpha$.

By explictly expressing the process which transforms the word-context co-occurrence matrix $M$ to the embedding matrix $E$, we can find a fact that the relative distances between words only changes within the scaling step. Considering vectors of all words are scaled the same at every dimension, it seems that the relative distances among words should be preserved to some extent. In fact, we have the following two theorems:

**Theorem 1.** *For any two words $w_i$ and $w_j$, if $\|M_{i,:} - M_{j,:}\|_2 \leq \delta$ ($\delta \geq 0$), then $\|E_{i,:} - E_{j,:}\|_2 \leq \sigma_d^{(\alpha-1)}\delta$.*

**Theorem 2.** *For any two words $w_i$ and $w_j$, if $\|E_{i,:} - E_{j,:}\|_2 \leq \delta$ ($\delta \geq 0$), then $\|M_{i,:} - M_{j,:}\|_2 \leq \sigma_1^{(1-\alpha)}\delta + 2(\sum_{k=d+1}^n \sigma_k^2)^{1/2}$.*

The formal proof of theorem 1 and theorem 2 is in appendix A.2 and A.3.

These two theorems describe how the relative distances (measured in Euclidean distance) among words change during the embedding transformation. Particularly, theorem 1 shows that if the distance between two words $w_i$ and $w_j$ in $M$ space is no more than $\delta$, their distance in the embedding space is also within a constraint. This bound is related with the singular value $\sigma_d$ and the parameter $\alpha$. So theorem 1 implies that the relative distances between words will not change dramatically during this transformation from $M$ to $E$ with suitable $\alpha$. Theorem 2 demonstrates that if we observe two words are embedded closely in $E$ ($\|E_{i,:} - E_{j,:}\|_2$ is small), we can infer these two words are also close in the original $M$ space. To sum up, the above two theorems indicate the neighborhood structure (relative distances to other words) for every word will be preserve to some extent during the transformation from $M$ to $E$. In other words, matrix $E$ inherits the neighborhood structure that exists in $M$. As the result, $E$ will inherit the properties in $M$ that are related with the relative distances among words. Now we can answer the question raised in the introduction: *What is the inner mechanism resulting in the good properties of word embedding?* In fact, we will show that these properties are the result of inheritance of neighborhood structure.

## 3.2 THE INNER MECHANISM RESULTING IN THE NICE PROPERTIES IN THE EMBEDDING

Let us focus on two important properties in $E$: (1) words with similar semantic meanings are likely to be embedded closely in the embedding space; (2) the word embedding tends to exhibit the analogy structure, e.g., "woman is to queen as man is to king". Note that these two properties are related with the relative distances among words. So, considering the previous analysis which shows matrix $E$ inherits the neighborhood structure existing in $M$, these two properties in $E$ also inherits from $M$ as the result.

**Words with similar semantic meanings are embedded closely.** Harris (1954) indicated that *if two words have almost identical environments we say that they are synonyms.*. In other words, two words will have similar neighbors if they have similar semantic meanings. It means the corresponding rows in the word-context co-occurrence matrix $M$ should be close for these words. So the words with similar semantic meanings are also close in the $M$ space which has been empirically verified in Levy et al. (2015). Remember that theorem 1 shows if the distance between two words in $M$ is close (within a small range $\delta$), these two words in $E$ will also be close due to this low rank transformation process from $M$ to $E$. So the low rank transformation in fact makes the embedding $E$ inherit this property existing in the original word-context matrix $M$.

**The analogy structure in the embedding.** Different from word similarity which is about the relative distance between two words (e.g., *king* and *queen*), the analogy structure is related with the relative distance between two pairs (e.g., *(king, queen)* and *(man, woman)*). So let us first examine how the relative distance between two pairs change during the embedding process.

**Theorem 3.** *Let $M_{\gamma_1,:}$, $M_{\gamma_2,:}$, $M_{\beta_1,:}$ and $M_{\beta_2,:}$ be four rows in $M$ matrix. Denote $M_\gamma = (M_{\gamma_1,:} - M_{\gamma_2,:})$ and $M_\beta = (M_{\beta_1,:} - M_{\beta_2,:})$. If $\|M_{\gamma,:} - M_{\beta,:}\|_2 \leq \delta$ ($\delta \geq 0$), then $\|E_{\gamma,:} - E_{\beta,:}\|_2 \leq \sigma_d^{(\alpha-1)}\delta$.*

**Theorem 4.** *Let $M_{\gamma_1,:}$, $M_{\gamma_2,:}$, $M_{\beta_1,:}$ and $M_{\beta_2,:}$ be four rows in $M$ matrix. Denote $M_\gamma = (M_{\gamma_1,:} - M_{\gamma_2,:})$ and $M_\beta = (M_{\beta_1,:} - M_{\beta_2,:})$. If $\|E_{\gamma,:} - E_{\beta,:}\|_2 \leq \delta$ ($\delta \geq 0$), then $\|M_{\gamma,:} - M_{\beta,:}\|_2 \leq \sigma_1^{(1-\alpha)}\delta + 4(\sum_{k=d+1}^n \sigma_k^2)^{1/2}$.*

The proof of theorem 3 and theorem 4 is in appendix A.4 and A.5, which is quite similar to the proof of theorem 1 and theorem 2.

Theorems 3 and 4 show that the relative distance between two pairs will not change dramatically during the embedding process either. In other words, the relative distance between pairs of words will be preserved during the embedding transformation from $M$ to $E$. Furthermore, Levy & Goldberg (2014a) revealed a fact that the original word-context co-occurrence $M$ (e.g., $M$ can be the PPMI matrix)[2] also exhibits the analogy structure, such that $M_{king} - M_{queen} \approx M_{man} - M_{woman}$. So

---

[2]The word-context co-occurrence matrix $M$ is called *explicit representation* in Levy & Goldberg (2014a)

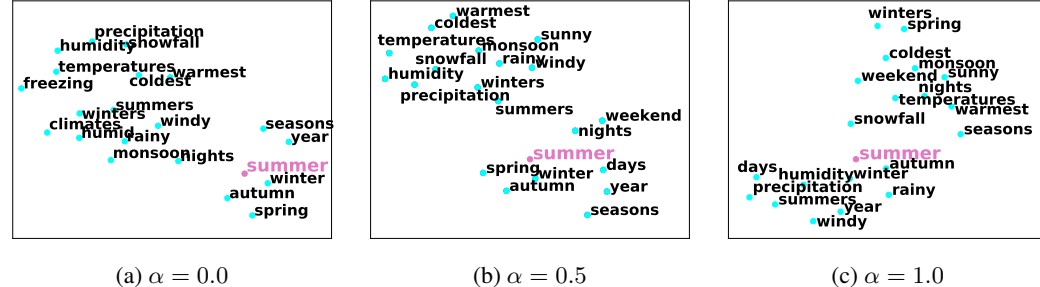

(a) $\alpha = 0.0$      (b) $\alpha = 0.5$      (c) $\alpha = 1.0$

Figure 1: Illustration of the local neighborhood structure in the embedding space $\boldsymbol{E} = \boldsymbol{U}_d \cdot \boldsymbol{\Sigma}_d^\alpha$ with different $\alpha$. From (a) to (c): t-SNE (Maaten & Hinton, 2008) visualization of the nearest 20 neighbors of an arbitrarily chosen word *summer* with $\alpha$ equals to 0.0, 0.5 and 1.0 respectively. More results corresponding other $\alpha$ can be founded in appendix A.9.

theorems 3 and 4 together with this fact indicate the analogy structure in $\boldsymbol{E}$ also inherits from $\boldsymbol{M}$ as the result of distance preserving in such low rank transformation process.

To summarize, we prove the bound about the change of relative distances of words during the embedding process. With appropriate $\alpha$, the neighborhood structure (relative distance) of every word will not change much during the embedding process. So the embedding $\boldsymbol{E}$ inherits the word similarity and the analogy structure from $\boldsymbol{M}$. We also theoretically explain how the parameter $\alpha$ influences the word embedding by controling the change of relative distances among words (the bounds in theorem 1-4). So a consideration is that we can optimize these bounds by choosing suitable $\alpha$. We show the derivation of finding the suitable $\alpha$ with respect to the bounds in appendix A.7. It shows the suitable $\alpha$ depends on $\sigma_1$ and $\sigma_d$, together with how to weight the two bounds in theorem 1 (3) and theorem 2 (4). However, the optimization over the bound only gives a rough estimation of the possible range of optimal $\alpha$. In the next section, we optimize $\alpha$ by directly minimizing the change of neighborhood of words instead of optimizing over the bound.

## 4    IMPROVEMENT: WORD EMBEDDING PRESERVING THE DISTANCE STRUCTURE BETTER

In previous section, we reveal the inner mechanism why the embedding exhibits such nice properties. Namely, the embedding is in fact a low rank transformation from $\boldsymbol{M}$ to $\boldsymbol{E}$ and the relative distances between words are preserved during such process. As a result, the embedding matrix $\boldsymbol{E}$ *inherits* these nice properties existing in the original word-context co-occurrence matrix $\boldsymbol{M}$ (e.g., the PMI matrix). Particularly, theorem 1, 2, 3 and 4 show that the parameter $\alpha$ has an important influence on how the relative distances between words change. For example, we arbitrarily select a word *summer* in the corpus[3] and plot its nearest 20 neighbors with $\alpha$ equal to 0.0, 0.5 and 1.0 respectively. Figure 1 shows that the neighborhood of *summer* changes with different $\alpha$. This observation gives us an inspiration that we can learn the embedding $\boldsymbol{E} = \boldsymbol{U}_d \cdot \boldsymbol{\Sigma}_d^\alpha$ by selecting a good $\alpha$, thus preserving the relative distances among words during the embedding transformation process.

**Preserving neighborhood structure better by choosing optimal $\alpha$ for embedding $\boldsymbol{E} = \boldsymbol{U}_d \cdot \boldsymbol{\Sigma}_d^\alpha$.** Let $g^{\mathcal{X}}(w)$ represent the neighbors of word $w$ in $\mathcal{X}$ space, e.g., $\mathcal{X}$ can be the word-context co-occurrence matrix $\boldsymbol{M}$ or the embedding matrix $\boldsymbol{E}$. Let $l(\cdot, \cdot)$ be a correlation function measuring the similarity of two different neighborhoods. This measurement captures how much the neighborhood structure changes incurred by the embedding process. For example, if we consider the top $knn$ nearest neighbors as $g^{\mathcal{X}}(\cdot)$ and ignore the order of these nearest neighbors, $g^{\mathcal{X}}(w)$ can be viewed as a set and $l$ can be a similarity measurement defined on sets (e.g., Jaccard Similarity). Otherwise, if we consider the order of neighbors, $g^{\mathcal{X}}(w)$ can be a ranking list such that the neighbors are ranked

---
[3]We will introduce the corpus in section 5

by their relative distance to $w$, and $l$ now can be a measurement about the correlation between two rankings. We use $L$ to denote the aggregated similarity score over all words

$$L = \frac{1}{n} \sum_{w \in W} l(g^{\boldsymbol{M}}(w), g^{\boldsymbol{E}}(w)) \tag{3}$$

where $W$ is the word vocabulary to consider and its size is $n$. Based on previous analysis, we argue we can select an optimal $\alpha$ for embedding $\boldsymbol{E} = \boldsymbol{U}_d \cdot \boldsymbol{\Sigma}_d^\alpha$ to preserve the relative distances among words during the embedding transformation, such that

$$\alpha^\star = \arg\max_\alpha \frac{1}{n} \sum_{w \in W} l(g^{\boldsymbol{M}}(w), g^{\boldsymbol{E}}(w)) \tag{4}$$

**Leveraging the effect of $\alpha$ into the original `SGNS` network structure.** The above analysis is based on the view that we compute the embedding matrix $\boldsymbol{E}$ by SVD such that $\boldsymbol{E} = \boldsymbol{U}_d \cdot \boldsymbol{\Sigma}_d^\alpha$. We can also think about to incorporate the effect of $\alpha$ into the original `SGNS` neural network structure. The original `SGNS` implicitly performing a symmetric factorization, thus implying the $\alpha$ equal to 0.5. Here, we consider to leverage the influence of $\alpha$ directly into the `SGNS` structure. Consider the embedding $\boldsymbol{E} = \boldsymbol{U}_d \cdot \boldsymbol{\Sigma}_d^\alpha$ and the context $\boldsymbol{C} = \boldsymbol{\Sigma}_d^{(1-\alpha)} \cdot \boldsymbol{V}_d^T$ learnt by SVD, we have $\frac{\log(\|\boldsymbol{e}_i\|_2^2)}{\log(\|\boldsymbol{e}_i\|_2^2 \|\boldsymbol{c}_i\|_2^2)} = \alpha$ [4], where $\boldsymbol{e}_i$ and $\boldsymbol{c}_i$ are the word vector $w_i$ and the context vector of $w_i$ respectively. We put it as a regularization term in the original `SGNS` objective function to add the influence of different $\alpha$ to the embedding learnt by `SGNS`. The detailed derivation is in appendix A.6.

To summarize, the $\alpha$ parameter in the embedding matrix $\boldsymbol{E} = \boldsymbol{U}_d \cdot \boldsymbol{\Sigma}_d^\alpha$ influences how the relative distances between words change during the embedding process, thus influencing the quality of the learnt embedding matrix. We propose a method to choose the optimal $\alpha$ by preserving the relative distance among words during the embedding transformation. Besides, we come up with an idea to incorporate the effect of $\alpha$ into the original `SGNS` architecture. In the next section, we will test these ideas on real experiments.

## 5 EXPERIMENTS

In this section, we conduct experiments to verify our previous analysis. The first test is to verify that a good embedding does maximize the correlation defined in equation 3 which measures to which extent the neighborhood structure is preserved during the embedding transformation. The second is to test if the proposed method to incorporate $\alpha$ into original `SGNS` architecture really has improvements.

### 5.1 EXPERIMENT SETTINGS

**Corpus** We use the Text9 corpus (Mahoney, 2011) which is a standard benchmark used for various natural language tasks. We follow the same pre-processing steps in Levy et al. (2015).

**Word similarity task** The purpose of the experiments is to verify whether the method proposed in section 4 to find the optimal $\alpha$ matches the real case. We adapt the same experiment as in Levy et al. (2015) to test the word embedding on the word similarity and analogy structure. There are six datasets acting as the ground truth for the word similarity: *WordSim353* (Finkelstein et al., 2002) and its two subsets *WS Similarity* and *WS Relatedness* (Zesch et al., 2008; Agirre et al., 2009), *Bruni MEN* (Bruni et al., 2012), *Radinsky Mechanical Turk* (Radinsky et al., 2011) and *Luong Rare Words* (Luong et al., 2013). These datasets[5] contain word pairs together with human-assigned similarity scores. Same as Levy et al. (2015), word embedding is evaluated by the correlation between the similarity computed by the embedding vector and the human-assigned similarity score. The function to compute correlation is the Spearman's correlation (same as Levy et al. (2015)) Note

---

[4]Suppose we only consider the words for which $\|\boldsymbol{e}_i\|_2^2 \|\boldsymbol{c}_i\|_2^2 \neq 1$, to avoid the case that zero appears in the denominator.

[5]These datasets can be downloaded from `https://bitbucket.org/omerlevy/hyperwords`

that we discard the pairs in the test file if it contains a word that does not appear in the *text9* corpus. We use the word similarity task as an example for verifying the proposed method (equation 3 and equation 4) to learn the optimal $\alpha$ for word embedding $\boldsymbol{E} = \boldsymbol{U}_d \cdot \boldsymbol{\Sigma}_d^\alpha$, and the proposed method to improve SGNS.

The codes used for the SVD embedding is adapted from `https://bitbucket.org/omerlevy/hyperwords` and the codes used to learn the SGNS embedding is adapted from `https://github.com/theeluwin/pytorch-sgns`. We store the codes in a shared dropbox folder [6] for the double-blind review.

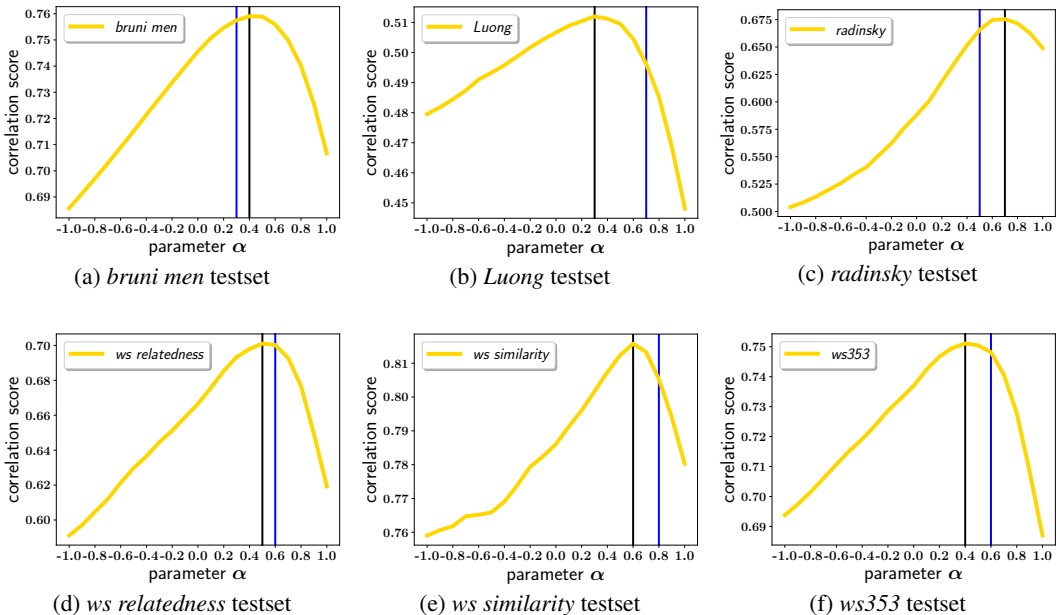

Figure 2: Compare the best $\alpha$ (black vertical line) in terms of the performance on the word similarity task and the best $\alpha^\star$ (blue vertical line) in terms of equation 4. $\alpha^\star$ is searched from $[-1, 1]$ with 0.1 as increment. From (a) to (e): y-axis is the correlation between the embedding score and the human assignend score. x-axis is different $\alpha$.

Table 1: The performance and similarity corresponding to $\alpha = 0$ (SVD embedding) and $\alpha = 0.5$ (symmetric embedding). Left is the peformance on every testset and right is the similarity computed by equation 3, considering the words appearing in that testset.

| $\alpha$ | bruni men | Luong | radinsky | ws relatedness | ws similarity | ws353 |
|---|---|---|---|---|---|---|
| 0.0 | 0.746 / 0.492 | 0.507 / 0.469 | 0.588 / 0.507 | 0.667 / 0.552 | 0.786 / 0.533 | 0.737 / 0.549 |
| 0.5 | 0.759 / 0.518 | 0.509 / 0.475 | 0.666 / 0.525 | 0.701 / 0.573 | 0.812 / 0.558 | 0.750 / 0.569 |

## 5.2 RESULTS

Firstly, let us verify that if the embedding $\boldsymbol{E} = \boldsymbol{U}_d \cdot \boldsymbol{\Sigma}_d^\alpha$ is good (measured by the performance on the word similarity testsets), the embedding $\boldsymbol{E}$ should preserve the relative distances between words better (measured by equation 3). We use the "vanilla" setting in Levy et al. (2015) to construct the positive PMI (PPMI) matrix as $\boldsymbol{M}$: window size is 2, no dynamic context window and no subsampling, the number of negative samples is 1 and the context distribution smoothing parameter is 1.0. For equation 3, we only consider the nearest neighbor for $g^\mathcal{X}$ and use Jaccard Similarity as the $l(\cdot, \cdot)$. Note that each testset only contains a subset of words in *text9*, and we also compute the overall correlation (the $L$ in equation 3) considering the words in the corresponding testset. The

---

[6]The dropbox link is `https://www.dropbox.com/sh/5d5j4pthcgzutdf/AABUvZPJpxUo8ugff1gQ7fIQa?dl=0`

result is in figure 1. Figure 1 shows that the performance (measured by these six word similarity groundtruth) and the proposed measurement are highly correlated, such that, for every testset, if $\alpha$ achieves high score word similarity test, the corresponding measurement in equation 3 is also high. More details can be found in appendix A.8. Besides we test whether the $\alpha^\star$ in equation 4 really corresponds to $\alpha$ that achieves the best performance on the word similarity task. We plot the performance of $\boldsymbol{E} = \boldsymbol{U} \cdot \boldsymbol{\Sigma}$ with different $\alpha$ in figure 2. The black vertical line corresponds to $\alpha$ that achieves best accuracy and the blue vertical line corresponds to the $\alpha^\star$ learnt by equation 4. We can see that $\alpha^\star$ is very near to the optimal point in almost all datasets. The only exception is the *Luong* testset, which contains many rare words. We infer the reason is that the corpus cannot effectively capture this testset because more than two thirds words in *Luong* testsets are not in the *text9* corpus.

Table 2: The results of incorporating $\alpha$ to the original SGNS.

| $\alpha$ | bruni men | Luong | radinsky | ws relatedness | ws similarity | ws353 |
|---|---|---|---|---|---|---|
| 0.3 | 0.70680 | 0.45004 | 0.67119 | 0.63262 | 0.77532 | 0.71247 |
| 0.5 | 0.70475 | 0.43613 | 0.67177 | 0.63733 | 0.77679 | 0.71309 |

Secondly, we test the effect that incorporating $\alpha$ into SGNS. Table 2 is the performance of the original SGNS with $\alpha$ equals to 0.3 and 0.5 .According to the result (the gold line) in figure 2, if we push the $\alpha$ from 0.5 to 0.3, the performance in the *Luong* testset should become better and the performance in all other testsets should drop. The results in table 2 is same with this inference. The only exception is the *bruni men* testset. We infer the reason is that the performances with $\alpha = 0.3$ and $\alpha = 0.5$ are very close (see gold line in figure 2 (a)).

## 6 RELATED WORK

**Explanation about the analogy structure.** Mikolov et al. (2013c) observed the analogy structure existing in the embedding created by a neural network. Levy & Goldberg (2014a) afterwards found the same property also exists in the original word-context co-occurrence matrix. Levy & Goldberg (2014a) infers *neural embedding process is not discovering novel patterns, but rather is doing a remarkable job at preserving the patterns inherent in the word-context co-occurrence matrix*. However, it is not clear how the neural embedding process preserves such property. Later, several explanations were been proposed to explain the reason (Arora et al., 2016; Gittens et al., 2017; Allen & Hospedales, 2019; Ethayarajh et al., 2018). For example, Gittens et al. (2017) and Allen & Hospedales (2019) referred to the *paraphrasing* to explain the analogy structure. Different from these previous studies, we analyze word embedding from the low rank transformation perspective and we reveal the inner mechanism how this property is inherited during the embedding transformation process.

**Explanation about the $\alpha$.** The influence of $\alpha$ on the quality of word embedding has been observed in (Caron, 2001; Turney, 2012; Bullinaria & Levy, 2012; Levy et al., 2015; Artetxe et al., 2018). These works *empirically* find that $\alpha$ has an important influence to the embedding and suggest this parameter should be tuned. However, these works do not provide theoretical explanation and no clear method has been provided to tune the $\alpha$. Yin & Shen (2018) is the only work we know which discusses the meaning of $\alpha$ from the pairwise inner product (PIP) loss perspective. But their findings have some contradiction to the real experiments in Caron (2001),Turney (2012), Bullinaria & Levy (2012), Levy et al. (2015) and Artetxe et al. (2018). Specifically, the analysis in Yin & Shen (2018) suggests small $\alpha$ is easier to result in over-fitting and lead to the performace drop, which implies that larger $\alpha$ is better. However, the real experiments in other papers show the performance also drops with very large $\alpha$. To summarize, we explain the meaning of $\alpha$ in the word embedding transformation which once was not theoretically clear. Futhermore, we propose a method to find the best $\alpha$ for word embedding.

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

## A  APPENDIX

### A.1  PROOF OF OBSERVATION 1

*Proof.* Firstly, we have

$$\boldsymbol{E} = \boldsymbol{M}_d \cdot \boldsymbol{\Sigma}_d^{\alpha}$$

$$= [\boldsymbol{U}_{:,1}, \boldsymbol{U}_{:,2}, \dots, \boldsymbol{U}_{:,d}] \cdot \begin{bmatrix} \sigma_1 & & \\ & \ddots & \\ & & \sigma_d \end{bmatrix}$$

$$= \left[ \sigma_1^{(\alpha)} \boldsymbol{U}_{:,1}, \sigma_2^{(\alpha)} \boldsymbol{U}_{:,2}, \dots, \sigma_d^{(\alpha)} \boldsymbol{U}_{:,d} \right]$$

Besides,

$$\boldsymbol{E}_{pse} = \boldsymbol{M} \cdot \boldsymbol{V} \cdot \boldsymbol{\Sigma}_{pse}^{(\alpha-1)}$$

$$= \boldsymbol{U} \cdot \boldsymbol{\Sigma} \cdot \boldsymbol{V}^T \cdot \boldsymbol{V} \cdot \boldsymbol{\Sigma}_{pse}^{(\alpha-1)}$$

$$= \boldsymbol{U} \cdot \boldsymbol{\Sigma} \cdot \boldsymbol{I} \cdot \boldsymbol{\Sigma}_{pse}^{(\alpha-1)} \ (\boldsymbol{V} \text{ is unitary matrix, so } \boldsymbol{V}^T \cdot \boldsymbol{V} = \boldsymbol{I})$$

$$= \boldsymbol{U} \cdot \boldsymbol{\Sigma} \cdot \boldsymbol{\Sigma}_{pse}^{(\alpha-1)}$$

$$= [\boldsymbol{U}_{:,1}, \boldsymbol{U}_{:,2}, \dots, \boldsymbol{U}_{:,n}] \cdot \begin{bmatrix} \sigma_1 & & & & & \\ & \ddots & & & & \\ & & \sigma_d & & & \\ & & & \sigma_{d+1} & & \\ & & & & \ddots & \\ & & & & & \sigma_n \end{bmatrix} \cdot \begin{bmatrix} \sigma_1^{(\alpha-1)} & & & & & \\ & \ddots & & & & \\ & & \sigma_d^{(\alpha-1)} & & & \\ & & & 0 & & \\ & & & & \ddots & \\ & & & & & 0 \end{bmatrix}$$

$$= [\boldsymbol{U}_{:,1}, \boldsymbol{U}_{:,2}, \dots, \boldsymbol{U}_{:,n}] \cdot \begin{bmatrix} \sigma_1^{(\alpha)} & & & & & \\ & \ddots & & & & \\ & & \sigma_d^{(\alpha)} & & & \\ & & & 0 & & \\ & & & & \ddots & \\ & & & & & 0 \end{bmatrix}$$

$$= [\sigma_1^{(\alpha)} \boldsymbol{U}_{:,1}, \sigma_2^{(\alpha)} \boldsymbol{U}_{:,2}, \dots, \sigma_d^{(\alpha)} \boldsymbol{U}_{:,d}, 0, \dots, 0]$$

So $\boldsymbol{E}$ corresponds to the first $d$ columns of $\boldsymbol{E}_{pse}$. ☐

### A.2  THE PROOF OF THEOREM 1

We need the following lemma for the proof of theorem 1.

**Lemma 1.** *Unitary matrix does not change the norm of a vector, such that*

$$\|\boldsymbol{U}\boldsymbol{a}\|_2 = \|\boldsymbol{a}\|_2$$

*where $\boldsymbol{U}$ is a unitary matrix and $\boldsymbol{a} \in \mathbb{R}^n$ is a n-dimension column vector.*

*Proof.* The norm can be expressed by using the Hermitian product, $(\cdot, \cdot)$

$$\|\boldsymbol{U}\boldsymbol{a}\|_2^2 = ((\boldsymbol{U}\boldsymbol{a}, \boldsymbol{U}\boldsymbol{a})) = (\boldsymbol{a}, \boldsymbol{U}^*\boldsymbol{U}\boldsymbol{a}) = (\boldsymbol{a}, \boldsymbol{a}) = \|\boldsymbol{a}\|_2^2$$

$$\Rightarrow \|\boldsymbol{U}\boldsymbol{a}\|_2 = \|\boldsymbol{a}\|_2$$

☐

Now we can prove the theorem 1. For any two words $w_i$ and $w_j$, if $\|\boldsymbol{M}_{i,:} - \boldsymbol{M}_{j,:}\|_2 \leq \delta$, then $\|\boldsymbol{E}_{i,:} - \boldsymbol{E}_{j,:}\|_2 \leq \left( \sigma_1^{(\alpha-1)} \right) \times \delta$.

*Proof.* Let $\tilde{E}$ denote the $E_{pse}$ defined in section 3. It is obvious that $\|E_{i,:} - E_{j,:}\|_2 = \left\|\tilde{E}_{i,:} - \tilde{E}_{j,:}\right\|_2$ since $E$ is the first $d$ columns of $\tilde{E}$ and the last $(n-d)$ dimensions in $\tilde{E}$ are zero.

Let $h = (h_1, h_2, \ldots, h_n)$ denote the $(M_{i,:} - M_{j,:})V$. We have $\|h\|_2 = \|(M_{i,:} - M_{j,:})V\|_2$. According to lemma 1, $\|(M_{i,:} - M_{j,:})V\|_2 = \|(M_{i,:} - M_{j,:})\|_2$. So we have $\|h\|_2 = \|(M_{i,:} - M_{j,:})V\|_2 = \|(M_{i,:} - M_{j,:})\|_2 \le \delta$.

$$
\begin{aligned}
&\|E_{i,:} - E_{j,:}\|_2 \\
&= \left\|\tilde{E}_{i,:} - \tilde{E}_{j,:}\right\|_2 \\
&= \left\|M_{i,:} \cdot V \cdot \Sigma_{pse}^{(\alpha-1)} - M_{j,:} \cdot V \cdot \Sigma_{pse}^{(\alpha-1)}\right\|_2 \\
&= \left\|(M_{i,:} \cdot V - M_{j,:} \cdot V) \cdot \Sigma_{pse}^{(\alpha-1)}\right\|_2 \\
&= \left\|h \cdot \Sigma_{pse}^{(\alpha-1)}\right\|_2 \\
&= ((\sigma_1^{(\alpha-1)} h_1)^2 + (\sigma_2^{(\alpha-1)} h_2)^2 + \ldots + (\sigma_d^{(\alpha-1)} h_d)^2 + 0 + \ldots + 0)^{1/2} \\
&\le (\sigma_d^{2(\alpha-1)} \cdot (h_1^2 + h_2^2 + \ldots + h_d^2))^{1/2} \text{ (note } \sigma_d \text{ is the smallest singular value)} \\
&\le (\sigma_d^{2(\alpha-1)} \cdot \|h\|_2^2)^{1/2} \le \sigma_d^{(\alpha-1)} \cdot \delta
\end{aligned}
$$

$\square$

## A.3 THE PROOF OF THEOREM 2

*Proof.* Let $M_{pse}$ be $U\Sigma_{pse}V^T$. It is easy to show that $M_{pse}$ equals to $U_d \cdot \Sigma_d \cdot V_d^T$.

Let $\tilde{M}$ denote $M_{pse}$. We first show that $\left\|M_{i,:} - \tilde{M}_{i,:}\right\|_2^2$ (for every $i$) is constrained by the singular values of $M$. Firstly, we have

$$
\left\|M_{i,:} - \tilde{M}_{i,:}\right\|_2 = (\sum_j (M_{i,j} - \tilde{M}_{i,j})^2)^{1/2} \le (\sum_k \sum_j (M_{k,j} - \tilde{M}_{k,j})^2)^{1/2} = \left\|M - \tilde{M}\right\|_2
$$

So we have

$$
\begin{aligned}
&\left\|M_{i,:} - \tilde{M}_{i,:}\right\|_2 \\
&\le \left\|M - \tilde{M}\right\|_2 \\
&= \left\|U\Sigma V^T - U\Sigma_{pse}V^T\right\|_2 \\
&= \left\|U(\Sigma - \Sigma_{pse})V^T\right\|_2 \\
&= \|(\Sigma - \Sigma_{pse})\|_2 \text{ (lemma 1)} \\
&= (\sigma_{d+1}^2 + \sigma_{d+2}^2 + \ldots + \sigma_n^2)^{1/2}
\end{aligned}
$$

Then we show if $\|E_{i,:} - E_{j,:}\|_2 \le \delta$, then $\left\|\tilde{M}_{i,:} - \tilde{M}_{j,:}\right\|_2 \le \delta\sigma_1^{(1-\alpha)}$.

$$
\begin{aligned}
&\left\|\tilde{M}_{i,:} - \tilde{M}_{j,:}\right\|_2 \\
&= \left\|(E_{i,:} - E_{j,:}) \cdot \Sigma_d^{(1-\alpha)} \cdot V_d^T\right\|_2 \\
&= \left\|(E_{i,:} - E_{j,:}) \cdot \Sigma_d^{(1-\alpha)}\right\|_2 \text{ (lemma 1)}
\end{aligned}
$$

Let $\boldsymbol{h} = (h_1, h_2 \ldots, h_n)$ denote $\boldsymbol{E}_{i,:} - \boldsymbol{E}_{j,:}$ here. We have $\|\boldsymbol{h}\|_2 = \|\boldsymbol{E}_{i,:} - \boldsymbol{E}_{j,:}\|_2 \le \delta$. So the above equation becomes

$$
\begin{aligned}
&\left\| (\boldsymbol{E}_{i,:} - \boldsymbol{E}_{j,:}) \cdot \boldsymbol{\Sigma}_d^{(1-\alpha)} \right\|_2 \\
=& \left\| (h_1, h_2 \ldots, h_n) \cdot \boldsymbol{\Sigma}_d^{(1-\alpha)} \right\|_2 \\
=& (h_1^2 \sigma_1^{2(1-\alpha)} + h_2^2 \sigma_2^{2(1-\alpha)} + \ldots + h_d^2 \sigma_d^{2(1-\alpha)})^{1/2} \\
\le& ((h_1^2 + h_2^2 + \ldots + h_d^2) \sigma_1^{2(1-\alpha)})^{1/2} \\
\le& \|\boldsymbol{h}\|_2 \, \sigma_1^{(1-\alpha)} \\
\le& \delta \sigma_1^{(1-\alpha)}
\end{aligned}
$$

With above preparations, we can now prove theorem 2.

$$
\begin{aligned}
&\|\boldsymbol{M}_{i,:} - \boldsymbol{M}_{j,:}\|_2 \\
=& \left\| (\boldsymbol{M}_{i,:} - \tilde{\boldsymbol{M}}_{i,:}) - (\boldsymbol{M}_{j,:} - \tilde{\boldsymbol{M}}_{i,:}) \right\|_2 \\
\le& \left\| (\boldsymbol{M}_{i,:} - \tilde{\boldsymbol{M}}_{i,:}) \right\|_2 + \left\| -(\boldsymbol{M}_{j,:} - \tilde{\boldsymbol{M}}_{i,:}) \right\|_2 \\
=& \left\| (\boldsymbol{M}_{i,:} - \tilde{\boldsymbol{M}}_{i,:}) \right\|_2 + \left\| (\boldsymbol{M}_{j,:} - \tilde{\boldsymbol{M}}_{j,:}) - (\tilde{\boldsymbol{M}}_{i,:} - \tilde{\boldsymbol{M}}_{j,:}) \right\|_2 \\
=& \underbrace{\left\| (\boldsymbol{M}_{i,:} - \tilde{\boldsymbol{M}}_{i,:}) \right\|_2}_{\le (\sum_{k=d+1}^n \sigma_k^2)^{1/2}} + \underbrace{\left\| (\boldsymbol{M}_{j,:} - \tilde{\boldsymbol{M}}_{j,:}) \right\|_2}_{\le (\sum_{k=d+1}^n \sigma_k^2)^{1/2}} + \underbrace{\left\| -(\tilde{\boldsymbol{M}}_{i,:} - \tilde{\boldsymbol{M}}_{j,:}) \right\|_2}_{\le \delta \sigma_1^{(1-\alpha)}} \\
\le& \delta \sigma_1^{(1-\alpha)} + 2 \Big( \sum_{k=d+1}^n \sigma_k^2 \Big)^{1/2}
\end{aligned}
$$

$\square$

### A.4 The Proof of Theorem 3

*Proof.* Similar to the proof in section A.2, let $\tilde{\boldsymbol{E}}$ denote the $\boldsymbol{E}_{pse}$ and $\boldsymbol{h} = (h_1, h_2 \ldots, h)$ denote the $((\boldsymbol{M}_{\gamma_1,:} - \boldsymbol{M}_{\gamma_2,:}) - (\boldsymbol{M}_{\beta_1,:} - \boldsymbol{M}_{\beta_2,:}))\boldsymbol{V}$. We have $\|\boldsymbol{h}\|_2 = \|((\boldsymbol{M}_{\gamma_1,:} - \boldsymbol{M}_{\gamma_2,:}) - (\boldsymbol{M}_{\beta_1,:} - \boldsymbol{M}_{\beta_2,:}))\boldsymbol{V}\|_2 = \|((\boldsymbol{M}_{\gamma_1,:} - \boldsymbol{M}_{\gamma_2,:}) - (\boldsymbol{M}_{\beta_1,:} - \boldsymbol{M}_{\beta_2,:}))\|_2 \le \delta$. Besides, $\|(\boldsymbol{E}_{\gamma_1,:} - \boldsymbol{E}_{\gamma_2,:}) - (\boldsymbol{E}_{\beta_1,:} - \boldsymbol{E}_{\beta_2,:})\|_2 = \left\| (\tilde{\boldsymbol{E}}_{\gamma_1,:} - \tilde{\boldsymbol{E}}_{\gamma_2,:}) - (\tilde{\boldsymbol{E}}_{\beta_1,:} - \tilde{\boldsymbol{E}}_{\beta_2,:}) \right\|_2$ since $\boldsymbol{E}$ is the first $d$ columns of $\tilde{\boldsymbol{E}}$ and the last $(n-d)$ dimensions in $\tilde{\boldsymbol{E}}$ are zero.

$$
\begin{aligned}
&\|(\boldsymbol{E}_{\gamma_1,:} - \boldsymbol{E}_{\gamma_2,:}) - (\boldsymbol{E}_{\beta_1,:} - \boldsymbol{E}_{\beta_2,:})\|_2 \\
=& \left\| (\tilde{\boldsymbol{E}}_{\gamma_1,:} - \tilde{\boldsymbol{E}}_{\gamma_2,:}) - (\tilde{\boldsymbol{E}}_{\beta_1,:} - \tilde{\boldsymbol{E}}_{\beta_2,:}) \right\|_2 \\
=& \left\| (\boldsymbol{M}_{\gamma_1,:} - \boldsymbol{M}_{\gamma_2,:}) \cdot \boldsymbol{V} \cdot \boldsymbol{\Sigma}_{pse}^{(\alpha-1)} - (\boldsymbol{M}_{\beta_1,:} - \boldsymbol{M}_{\beta_2,:}) \cdot \boldsymbol{V} \cdot \boldsymbol{\Sigma}_{pse}^{(\alpha-1)} \right\|_2 \\
=& \left\| ((\boldsymbol{M}_{\gamma_1,:} - \boldsymbol{M}_{\gamma_2,:}) - (\boldsymbol{M}_{\beta_1,:} - \boldsymbol{M}_{\beta_2,:})) \cdot \boldsymbol{V} \cdot \boldsymbol{\Sigma}_{pse}^{(\alpha-1)} \right\|_2 \\
=& \left\| \boldsymbol{h} \cdot \boldsymbol{\Sigma}_{pse}^{(\alpha-1)} \right\|_2 \\
=& ((\sigma_1^{(\alpha-1)} h_1)^2 + (\sigma_2^{(\alpha-1)} h_2)^2 + \ldots + (\sigma_d^{(\alpha-1)} h_d)^2 + 0 + \ldots + 0)^{1/2} \\
\le& \sigma_d^{(\alpha-1)} \cdot (h_1^2 + h_2^2 + \ldots + h_d^2)^{1/2} \text{ (note } \sigma_d \text{ is the smallest singular value)} \\
\le& \sigma_d^{(\alpha-1)} \cdot \|\boldsymbol{h}\|_2 \le \sigma_d^{(\alpha-1)} \cdot \delta
\end{aligned}
$$

$\square$

## A.5 THE PROOF OF THEOREM 4

The proof is quite similar to the proof in section A.3. Let $M_{pse}$ be $U\Sigma_{pse}V^T$. According to section A.3, we know $M_{pse}$ equals to $\tilde{U}_d \cdot \Sigma_d \cdot V_d^t$.

Let $\tilde{M}$ denote $M_{pse}$. According to section A.3, we have $\left\|M_{i,:} - \tilde{M}_{i,:}\right\|_2 \leq (\sigma_{d+1}^2 + \sigma_{d+2}^2 + \ldots + \sigma_n^2)^{1/2}$ (for every $i$).

Then with the same process in section A.3, we show if $\|(E_{\gamma_1,:} - E_{\gamma_2,:}) - (E_{\beta_1,:} - E_{\beta_2,:})\|_2 \leq \delta$, then $\left\|((\tilde{M}_{\gamma_1,:} - \tilde{M}_{\gamma_2,:}) - (\tilde{M}_{\beta_1,:} - \tilde{M}_{\beta_2,:}))\right\|_2 \leq \delta\sigma_1^{(1-\alpha)}$.

$$
\begin{aligned}
&\left\|((\tilde{M}_{\gamma_1,:} - \tilde{M}_{\gamma_2,:}) - (\tilde{M}_{\beta_1,:} - \tilde{M}_{\beta_2,:}))\right\|_2 \\
&= \left\|((E_{\gamma_1,:} - E_{\gamma_2,:}) - (E_{\beta_1,:} - E_{\beta_2,:})) \cdot \Sigma_d^{(1-\alpha)} \cdot V_d^T\right\|_2 \\
&= \left\|((E_{\gamma_1,:} - E_{\gamma_2,:}) - (E_{\beta_1,:} - E_{\beta_2,:})) \cdot \Sigma_d^{(1-\alpha)}\right\|_2 \quad \text{(lemma 1)}
\end{aligned}
$$

Let $h = (h_1, h_2, \ldots, h_n)$ denote $((E_{\gamma_1,:} - E_{\gamma_2,:}) - (E_{\beta_1,:} - E_{\beta_2,:}))$ here. We have $\|h\|_2 = \|((E_{\gamma_1,:} - E_{\gamma_2,:}) - (E_{\beta_1,:} - E_{\beta_2,:}))\|_2 \leq \delta$. So the above equation becomes

$$
\begin{aligned}
&\left\|((E_{\gamma_1,:} - E_{\gamma_2,:}) - (E_{\beta_1,:} - E_{\beta_2,:})) \cdot \Sigma_d^{(1-\alpha)}\right\|_2 \\
&= \left\|(h_1, h_2, \ldots, h_n) \cdot \Sigma_d^{(1-\alpha)}\right\|_2 \\
&= (h_1^2\sigma_1^{2(1-\alpha)} + h_2^2\sigma_2^{2(1-\alpha)} + \ldots + h_d^2\sigma_d^{2(1-\alpha)})^{1/2} \\
&\leq (h_1^2 + h_2^2 + \ldots + h_d^2)^{1/2}\sigma_1^{(1-\alpha)} \\
&= \|h\|_2 \, \sigma_1^{(1-\alpha)} \\
&= \delta\sigma_1^{(1-\alpha)}
\end{aligned}
$$

With above preparations, we can now prove theorem 4.

*Proof.*

$$
\begin{aligned}
&\|M_{\gamma,:} - M_{\beta,:}\|_2 \\
&= \left\|(M_{\gamma,:} - \tilde{M}_{\gamma,:}) - (M_{\beta,:} - \tilde{M}_{\gamma,:})\right\|_2 \\
&\leq \left\|(M_{\gamma,:} - \tilde{M}_{\gamma,:})\right\|_2 + \left\|-(M_{\beta,:} - \tilde{M}_{\gamma,:})\right\|_2 \\
&= \left\|(M_{\gamma,:} - \tilde{M}_{\gamma,:})\right\|_2 + \left\|(M_{\beta,:} - \tilde{M}_{\beta,:}) - (\tilde{M}_{\gamma,:} - \tilde{M}_{\beta,:})\right\|_2 \\
&\leq \left\|(M_{\gamma,:} - \tilde{M}_{\gamma,:})\right\|_2 + \left\|(M_{\beta,:} - \tilde{M}_{\beta,:})\right\|_2 + \left\|-(\tilde{M}_{\gamma,:} - \tilde{M}_{\beta,:})\right\|_2 \\
&= \left\|(M_{\gamma_1,:} - \tilde{M}_{\gamma_1,:}) - (M_{\gamma_2,:} - \tilde{M}_{\gamma_2,:})\right\|_2 + \left\|(M_{\beta_1,:} - \tilde{M}_{\beta_2,:}) - (M_{\beta_2,:} - \tilde{M}_{\beta_2,:})\right\|_2 + \left\|-(\tilde{M}_{\gamma,:} - \tilde{M}_{\beta,:})\right\|_2 \\
&\leq \underbrace{\left\|(M_{\gamma_1,:} - \tilde{M}_{\gamma_1,:})\right\|_2}_{\leq(\sum_{k=d+1}^n \sigma_k^2)^{1/2}} + \underbrace{\left\|-(M_{\gamma_2,:} - \tilde{M}_{\gamma_2,:})\right\|_2}_{\leq(\sum_{k=d+1}^n \sigma_k^2)^{1/2}} + \underbrace{\left\|(M_{\beta_1,:} - \tilde{M}_{\beta_2,:})\right\|_2}_{\leq(\sum_{k=d+1}^n \sigma_k^2)^{1/2}} + \underbrace{\left\|-(M_{\beta_2,:} - \tilde{M}_{\beta_2,:})\right\|_2}_{\leq(\sum_{k=d+1}^n \sigma_k^2)^{1/2}} + \underbrace{\left\|-(\tilde{M}_{\gamma,:} - \tilde{M}_{\beta,:})\right\|_2}_{\leq\delta\sigma_1^{(1-\alpha)}} \\
&= \delta\sigma_1^{(1-\alpha)} + 4(\sum_{k=d+1}^n \sigma_k^2)^{1/2}
\end{aligned}
$$

$\square$

## A.6 THE PROOF OF ALPHA REGULARIZATION

*Proof.* Firstly, we have

$$
\begin{aligned}
\boldsymbol{E} &= \boldsymbol{U}_d \cdot \boldsymbol{\Sigma}_d^{\alpha} \\
&= [\lambda_1^{\alpha} \boldsymbol{u}_1, \lambda_2^{\alpha} \boldsymbol{u}_2, ..., \lambda_d^{\alpha} \boldsymbol{u}_d] \\
\boldsymbol{C}^T &= \boldsymbol{\Sigma}_d^{1-\alpha} \cdot \boldsymbol{V}_d^T \\
&= \begin{bmatrix} \lambda_1^{1-\alpha} \boldsymbol{v}_1^T \\ \lambda_2^{1-\alpha} \boldsymbol{v}_2^T \\ \vdots \\ \lambda_d^{1-\alpha} \boldsymbol{v}_d^T \end{bmatrix}
\end{aligned}
$$

Since $\boldsymbol{U}$ and $\boldsymbol{V}$ is orthogonal, we have

$$
\boldsymbol{e}_i^T \boldsymbol{e}_j = \begin{cases} \lambda_i^{2\alpha}, & i = j \\ 0, & i \neq j \end{cases}
$$

$$
\boldsymbol{c}_i^T \boldsymbol{c}_j = \begin{cases} \lambda_i^{2-2\alpha}, & i = j \\ 0, & i \neq j \end{cases}
$$

Then, if $(\|\boldsymbol{e}_i\|_2^2 \cdot \|\boldsymbol{c}_i\|_2^2) \neq 1$, we have

$$
\frac{\log(\|\boldsymbol{e}_i\|_2^2)}{\log(\|\boldsymbol{e}_i\|_2^2 \cdot \|\boldsymbol{c}_i\|_2^2)} = \alpha
$$

$\square$

## A.7 THE OPTIMAL $\alpha$ IN TERMS OF MINIMIZING THE WEIGHTED SUMMATION OF THE BOUNDS IN THEOREM 1 AND 2

We assume that the optimal $\alpha$ should minimize the outcome bounds in both Theorem 1 ($B_1$) and 2 ($B_2$) [7]. The reason is that if two words are close in the original $\boldsymbol{M}$ space, these two words should also be close in the $\boldsymbol{E}$ space because we do not want the relative distances of words change dramatically, and vice versa. Thus we derive the optimal $\alpha$ by minimizing the weighted summation of the bounds

$$
B = \lambda B_1 + (1 - \lambda) B_2
$$

$$
B_1 = \begin{cases} \sigma_d^{(\alpha-1)} \delta, & \alpha \leq 1 \\ \sigma_1^{(\alpha-1)} \delta, & \alpha \geq 1 \end{cases}
$$

$$
B_2 = \begin{cases} \sigma_1^{(1-\alpha)} \delta, & \alpha \leq 1 \\ \sigma_d^{(1-\alpha)} \delta, & \alpha \geq 1 \end{cases}
$$

where $\lambda$ is the weight coefficient in the summation.

We derive the optimal $\alpha$ in three cases, i.e. Case 1: $\sigma_1 \geq \sigma_d \geq 1$, Case 2: $\sigma_1 \geq 1 \geq \sigma_d$, Case 3: $\sigma_d \leq \sigma_1 \leq 1$.

For Case 1, when $\alpha \geq 1$, we have

$$
B = \lambda \sigma_1^{\alpha-1} \delta + (1 - \lambda) \sigma_d^{1-\alpha} \delta
$$

$$
\frac{\partial B}{\partial \alpha} = \lambda \delta \sigma_1^{\alpha-1} \ln \sigma_1 - (1 - \lambda) \delta \sigma_d^{1-\alpha} \ln \sigma_d
$$

Easy to prove that $B$ is a convex function, so we derive $\alpha$ by solving the equation $\frac{\partial B}{\partial \alpha} = 0$, for simplicity, we use $k = \frac{1-\lambda}{\lambda}$,

$$
\frac{\partial B}{\partial \alpha} = 0 \Rightarrow \lambda \delta \sigma_1^{\alpha-1} \ln \sigma_1 = (1 - \lambda) \delta \sigma_d^{1-\alpha} \ln \sigma_d
$$

$$
\alpha = 1 + \frac{\ln \frac{k \ln \sigma_d}{\ln \sigma_1}}{\ln (\sigma_1 \sigma_d)}
$$

---

[7]The optimal $\alpha$ with respect to theorem 3 and theorem 4 can be derived in the same way.

Similarly, when $\alpha \leq 1$, we have

$$B = \lambda \sigma_d^{\alpha-1}\delta + (1-\lambda)\sigma_1^{1-\alpha}\delta$$

$$\frac{\partial B}{\partial \alpha} = 0 \Rightarrow \alpha = 1 + \frac{\ln \frac{k \ln \sigma_1}{\ln \sigma_d}}{\ln (\sigma_1 \sigma_d)}$$

Then we have the optimal $\alpha$ in terms of k as

$$\alpha = \begin{cases} 1 + \frac{\ln \frac{k \ln \sigma_d}{\ln \sigma_1}}{\ln (\sigma_1 \sigma_d)} \geq 1, & k \geq \frac{\ln \sigma_1}{\ln \sigma_d} \\ 1, & \frac{\ln \sigma_d}{\ln \sigma_1} \leq k \leq \frac{\ln \sigma_1}{\ln \sigma_d} \\ 1 + \frac{\ln \frac{k \ln \sigma_1}{\ln \sigma_d}}{\ln (\sigma_1 \sigma_d)} \leq 1, & k \leq \frac{\ln \sigma_d}{\ln \sigma_1} \end{cases}$$

For Case 2, from the formulation of $B$, we can easily derive the optimal $\alpha$ is 1.

For Case 3, similar with the derivation of Case 1, we have the optimal $\alpha$ in terms of k as,

$$\alpha = \begin{cases} 1 + \frac{\ln \frac{k \ln \sigma_1^{-1}}{\ln \sigma_d^{-1}}}{\ln (\sigma_1 \sigma_d)} \leq 1, & k \geq \frac{\ln \sigma_d^{-1}}{\ln \sigma_1^{-1}} \\ 1, & \frac{\ln \sigma_1^{-1}}{\ln \sigma_d^{-1}} \leq k \leq \frac{\ln \sigma_d^{-1}}{\ln \sigma_1^{-1}} \\ 1 + \frac{\ln \frac{k \ln \sigma_d^{-1}}{\ln \sigma_1^{-1}}}{\ln (\sigma_1 \sigma_d)} \geq 1, & k \leq \frac{\ln \sigma_1^{-1}}{\ln \sigma_d^{-1}} \end{cases}$$

A.8 PERFORMANCE ON WORD SIMILARITY TASK AND THE CORRELATION SCORE DEFINED IN EQUATION 3 WITH DIFFERENT $\alpha$

Table 3: The performance and similarity corresponding to different $\alpha$. Left is the peformance on every testset and right is the similarity computed by equation 3, considering the words appearing in that testset.

| $\alpha$ | bruni men | Luong | radinsky | ws relatedness | ws similarity | ws353 |
|---|---|---|---|---|---|---|
| $-1.0$ | 0.686 / 0.492 | 0.480 / 0.457 | 0.504 / 0.487 | 0.591 / 0.532 | 0.759 / 0.522 | 0.694 / 0.535 |
| $-0.9$ | 0.691 / 0.490 | 0.482 / 0.458 | 0.508 / 0.495 | 0.597 / 0.532 | 0.761 / 0.522 | 0.697 / 0.535 |
| $-0.8$ | 0.697 / 0.489 | 0.484 / 0.459 | 0.513 / 0.493 | 0.605 / 0.538 | 0.762 / 0.526 | 0.701 / 0.539 |
| $-0.7$ | 0.703 / 0.490 | 0.487 / 0.460 | 0.520 / 0.501 | 0.612 / 0.538 | 0.765 / 0.526 | 0.706 / 0.539 |
| $-0.6$ | 0.709 / 0.488 | 0.491 / 0.462 | 0.526 / 0.499 | 0.621 / 0.538 | 0.765 / 0.522 | 0.711 / 0.537 |
| $-0.5$ | 0.715 / 0.488 | 0.493 / 0.462 | 0.534 / 0.499 | 0.630 / 0.538 | 0.766 / 0.518 | 0.715 / 0.535 |
| $-0.4$ | 0.721 / 0.488 | 0.496 / 0.462 | 0.540 / 0.501 | 0.637 / 0.544 | 0.769 / 0.526 | 0.719 / 0.539 |
| $-0.3$ | 0.728 / 0.489 | 0.499 / 0.460 | 0.551 / 0.503 | 0.645 / 0.544 | 0.774 / 0.526 | 0.724 / 0.539 |
| $-0.2$ | 0.734 / 0.492 | 0.502 / 0.467 | 0.562 / 0.509 | 0.651 / 0.547 | 0.779 / 0.529 | 0.729 / 0.544 |
| $-0.1$ | 0.740 / 0.489 | 0.504 / 0.466 | 0.576 / 0.509 | 0.659 / 0.549 | 0.782 / 0.533 | 0.733 / 0.546 |
| $0.0$ | 0.746 / 0.492 | 0.507 / 0.469 | 0.588 / 0.507 | 0.667 / 0.552 | 0.786 / 0.533 | 0.737 / 0.549 |
| $0.1$ | 0.751 / 0.497 | 0.509 / 0.469 | 0.601 / 0.509 | 0.676 / 0.558 | 0.791 / 0.536 | 0.742 / 0.553 |
| $0.2$ | 0.755 / 0.504 | 0.510 / 0.471 | 0.618 / 0.513 | 0.685 / 0.558 | 0.796 / 0.540 | 0.747 / 0.553 |
| $0.3$ | 0.758 / 0.520 | 0.512 / 0.473 | 0.635 / 0.521 | 0.693 / 0.570 | 0.802 / 0.555 | 0.750 / 0.563 |
| $0.4$ | 0.759 / 0.519 | 0.511 / 0.475 | 0.652 / 0.525 | 0.698 / 0.570 | 0.807 / 0.555 | 0.751 / 0.565 |
| $0.5$ | 0.759 / 0.518 | 0.509 / 0.475 | 0.666 / 0.525 | 0.701 / 0.573 | 0.812 / 0.558 | 0.750 / 0.569 |
| $0.6$ | 0.756 / 0.515 | 0.504 / 0.478 | 0.675 / 0.517 | 0.700 / 0.576 | 0.816 / 0.558 | 0.748 / 0.572 |
| $0.7$ | 0.750 / 0.518 | 0.496 / 0.481 | 0.675 / 0.517 | 0.693 / 0.567 | 0.813 / 0.562 | 0.741 / 0.565 |
| $0.8$ | 0.740 / 0.527 | 0.485 / 0.472 | 0.672 / 0.511 | 0.677 / 0.576 | 0.805 / 0.573 | 0.728 / 0.572 |
| $0.9$ | 0.726 / 0.516 | 0.468 / 0.465 | 0.663 / 0.517 | 0.649 / 0.567 | 0.794 / 0.555 | 0.708 / 0.558 |
| $1.0$ | 0.707 / 0.505 | 0.448 / 0.456 | 0.649 / 0.513 | 0.619 / 0.561 | 0.780 / 0.547 | 0.687 / 0.556 |

## A.9 THE NEIGHBORHOOD OF *summer* WITH DIFFERENT $\alpha$

|  | top 20 nearest words |
|---|---|
| $PMI$ | winter, olympics, autumn, during, and, spring, annual, season, in, at
night, year, on, weather, after, rainy, day, including, festival, seasonal |

| $\alpha$ | top 20 nearest words |
|---|---|
| -1.0 | winter, autumn, spring, rainy, monsoon, summers, winters, warmest, nights, coldest
seasons, precipitation, temperatures, humidity, year, snowfall, freezing, months, average, climates |
| -0.9 | winter, autumn, spring, rainy, monsoon, summers, winters, warmest, nights, coldest
seasons, precipitation, temperatures,humidity, snowfall, year, freezing, months, average, climates |
| -0.8 | winter, autumn, spring, rainy, monsoon, summers, winters, warmest, nights, coldest
seasons, precipitation, snowfall, temperatures, humidity, year, freezing, climates, months, average |
| -0.7 | winter, autumn, spring, rainy, summers, monsoon, winters, warmest, nights, coldest
seasons, precipitation, snowfall, humidity, temperatures, year, freezing, climates, **humid**, months |
| -0.6 | winter, autumn, spring, rainy, summers, monsoon, winters, warmest, nights, coldest
snowfall, precipitation, seasons, humidity, temperatures, year, freezing, climates, humid, months |
| -0.5 | winter, autumn, rainy, spring, summers, monsoon, winters, warmest, nights, coldest
snowfall, precipitation, seasons, humidity, temperatures, year, freezing, climates, humid, months |
| -0.4 | winter, autumn, rainy, spring, summers, monsoon, winters, warmest, nights, coldest
snowfall, precipitation, humidity, seasons, temperatures, year, freezing, climates, humid, months |
| -0.3 | winter, autumn, rainy, spring, summers, monsoon, winters, nights, warmest, coldest
snowfall, precipitation, humidity, temperatures, seasons, year, climates, freezing, humid, **windy** |
| -0.2 | winter, autumn, rainy, spring, summers, winters, monsoon, nights, warmest, coldest
snowfall, precipitation, humidity, temperatures, seasons, year, climates, humid, freezing, windy |
| -0.1 | winter, autumn, rainy, spring, summers, winters, monsoon, nights, warmest, coldest
snowfall, precipitation, humidity, temperatures, seasons, climates, year, humid, freezing, windy |
| 0.0 | winter, autumn, rainy, spring, summers, winters, monsoon, nights, warmest, coldest
snowfall, precipitation, humidity, temperatures, seasons, climates, humid, year, windy, freezing |
| 0.1 | winter, autumn, rainy, spring, summers, winters, monsoon, nights, warmest, coldest
snowfall, precipitation, temperatures, humidity, seasons, year, humid, climates, windy, freezing |
| 0.2 | winter, autumn, rainy, spring, summers, winters, monsoon, nights, warmest, coldest
snowfall, precipitation, temperatures, humidity, seasons, year, humid, climates, windy, **sunny** |
| 0.3 | winter, autumn, rainy, spring, summers, winters, monsoon, nights, warmest, snowfall
coldest, precipitation, temperatures, seasons, humidity, year, humid, windy, climates, sunny |
| 0.4 | winter, autumn, rainy, spring, summers, winters, nights, monsoon, snowfall, warmest
coldest, precipitation, seasons, temperatures, year, humidity, windy, humid, climates, **days** |
| 0.5 | winter, autumn, rainy, spring, summers, nights, winters, monsoon, snowfall, warmest
coldest, seasons, precipitation, year, days, temperatures, **weekend**, windy, **sunny**, humidity |
| 0.6 | winter, autumn, rainy, spring, nights, summers, winters, monsoon, snowfall, warmest
coldest, seasons, year, days, weekend, precipitation, **annual**, sunny, temperatures, **month** |
| 0.7 | winter, autumn, rainy, spring, nights, summers, winters, monsoon, snowfall, warmest
days, coldest, year, seasons, weekend, annual, **week**, sunny, month, **snow** |
| 0.8 | winter, autumn, spring, rainy, nights, winters, summers, days, monsoon, year
weekend, seasons, snowfall, annual, week, **night**, warmest, coldest, snow, month |
| 0.9 | winter, autumn, spring, nights, rainy, days, winters, summers, night, year
weekend, annual, week, seasons, monsoon, snow, **holiday**, **rain**, month, **day** |
| 1.0 | winter, autumn, spring, nights, rainy, days, night, week, annual, year
weekend, seasons, winters, snow, summers, day, holiday, rain, month, **weather** |