# OpenReview forum: "Word embedding re-examined: is the symmetrical factorization optimal?"
_ICLR.cc/2020/Conference — Reject_

### Official Review · AnonReviewer3 · 2019-10-21
**Official Blind Review #3**

**Rating:** 3

**Review:**

This paper explores the role of the implicit alpha parameter when learning word embeddings. More concretely, word embeddings work by either implicitly or explicitly factorizing a co-occurrence matrix, and the underlying parameter alpha controls how the singular values are weighted between the word and the context vectors. The authors provide theoretical insights on the role of alpha in relation with the original co-occurrence matrix, and propose a new method to find its optimal value.

I think that this is overall a solid work. The paper provides a new perspective in the workings of word embeddings that I find interesting, it is theoretically well-founded (although I did not check all derivations in detail), and the presentation is clear.

However, I think that the paper does a poor job in putting its contributions into context in relation to previous work. In particular, the role of alpha in word embeddings was already studied empirically by Artetxe et al. (CoNLL'18, https://www.aclweb.org/anthology/K18-1028.pdf) for both word analogy and word similarity tasks, to the extent that Figure 2 in both papers is showing the exact same curves. However, the authors do not even cite it. As acknowledged in the paper, other authors like Levy et al. (TACL'15) also observed that the value of alpha was important in their experiments.

I think that the right narrative for the paper should more in the line of "previous work showed that alpha behaves this and that way; we provide a theoretical explanation for this behavior, and derive a method to automatically find its optimal value". However, starting from the title ("word embedding re-examined: is the symmetrical factorization optimal?", when it was already known that it wasn't) and the abstract (where only the statement that "we propose a method to find the optimal alpha" corresponds to a novel contribution), the paper does a poor job in identifying and properly contextualizing its real contributions. More importantly, the paper does not try to establish any connection between the authors own theory and the empirical findings from previous work.

In terms of the actual content, the authors constantly claim that word2vec is performing a symmetric factorization (e.g. "the original word2vec is implicitly performing a symmetric factorization, thus implying the alpha equal to 0.5) as if it was something obvious or well-known. I might be missing something here, but I do not see why this is the case. Following your notation, let's say that word2vec is implicitly factorizing M = E*C^T, where E are the word embeddings and C are the context embeddings. One could multiply E with any arbitrary invertible matrix W, and C by the transpose of its inverse W^-T, which could be chosen to completely break any symmetry, yet the objective value of word2vec would not change at all, as (E*W)*(C*W^-T)^T = E*C^T. In other words, there is nothing in the training objective of word2vec that forces a symmetric factorization, and there is always an optimal solution with respect to this training objective that is arbitrarily asymmetric.

Another point that raises concerns to me is that the optimal value of alpha is determined by the vocabulary of the evaluation task. It would make sense if the optimal alpha depended on the nature of the task (e.g. syntactic vs semantic), but I do not have any intuition (nor do the authors provide) as of why the vocabulary would be anyhow relevant. More importantly, this does not seem generalizable beyond a few intrinsic tasks as, in the general case, one wants good embeddings for the full vocabulary. In either case, I think that this point deserves more attention in the paper.

I also find the experimental evaluation to be somewhat weak. In particular, the proposed theory focuses in two phenomena (word similarity and word analogy) as stated in the abstract itself, but the empirical evaluation is limited to the word similarity task.

Also, this is a minor detail and it did not influence my score, but I dislike that the authors use "word2vec" to refer to skip-gram with negative sampling throughout the paper. I would suggest to either use SGNS (which is quite standard) or simply skip-gram.

**Experience Assessment:**

I have published in this field for several years.

**Review Assessment: Checking Correctness Of Derivations And Theory:**

I assessed the sensibility of the derivations and theory.

**Review Assessment: Checking Correctness Of Experiments:**

I assessed the sensibility of the experiments.

**Review Assessment: Thoroughness In Paper Reading:**

I read the paper thoroughly.

---

> ### Author Response · Authors · 2019-11-15
> **Thanks sincerely for your review!**
>
> Thank you very much for the valuable comments. We agree on almost every point in your comment. And we revise the narrative of some key points according to your suggestion.
>
> 1. Firstly, we revise the term word2vec to SGNS in the paper:)
>
> 2. About the discussion on the parameter $\alpha$.
> Thanks for this comment. We totally agree and revised the narrative of the discussion about $\alpha$. Particularly, we are sorry we once missed this related work by Artetxe et al. (CoNLL'18) and we add it into the revised version.
>
> 3. About the claim that SGNS is performing a symmetric factorization.
> Our claim that SGNS is performing a symmetric factorization is based on the existing works. The evidence includes: (1) "the factorization achieved by SGNS’s training procedure is much more “symmetric”..." stated in section 3.3 of Levy et al. (TACL'15, https://www.aclweb.org/anthology/Q15-1016.pdf); (2) "In Levy and Goldberg [2014] where the authors explained the connection between skip-gram Word2Vec and matrix factorization, $\alpha$ is set to 0.5 to enforce symmetry" stated in section 2.2 of Yin and Shen (NeurIPS'18); "Levy and Goldberg [2014] showed that skip-gram Word2Vec’s objective is an implicit symmetric factorization of the Pointwise Mutual Information (PMI) matrix" stated in section 2.3 in Yin and Shen (NeurIPS'18); "For the popular skip-gram [Mikolov et al., 2013b] and GloVe [Pennington et al., 2014], $\alpha$ equals 0.5 as they are implicitly doing a symmetric factorization" stated in section 5.1 in Yin and Shen (NeurIPS'18, https://papers.nips.cc/paper/7368-on-the-dimensionality-of-word-embedding.pdf).
>
> However, we realized this point is arguable as you mentioned. Anyway, it is not the key point in our paper. The purpose of us to use this claim is to highlight the influence of the parameter $\alpha$. Besides,  if the asymmetric decomposition is better than the symmetric decomposition, we can explicitly add this asymmetry into the SGNS architecture. We change the narrative in the updated version.
>
>
> 4. About the experiments.
> Our intuition of tuning $\alpha$ according to the specific training dataset is that we think every training dataset is focusing on some particular words or words pairs. For example, the dataset "bruni_men.txt" are focus on the similarity of word pairs appears in this dataset, and "luong_rare.txt" may focus on the similarity of other word pairs. So, if the goal is to make the word embedding perform well for a specific dataset, we can particularly consider the relative distance of words (word similarity task) and the relative distance of pairs (e.g. ("France", "Germany") and ("Paris", "Berlin")). Or if the goal is to learn a general word embedding that is not designed for a specific task, we can consider all words instead of some particular words. Besides, if we care about the efficiency, we can sample some words instead of using all words.

---

### Official Review · AnonReviewer1 · 2019-10-22
**Official Blind Review #1**

**Rating:** 3

**Review:**


Summary:
=======
This paper provides a closer look at the well-studied problem of learning word embeddings. In particular, it looks at the set of embedding methods that explicitly or implicitly perform a matrix factorization and tries to understand why the word embeddings exhibit analogy structure and why words that are semantically similar get embedded close together. The mechanism it comes up with has to do with the alpha parameter that represents the powers of singular values of the matrix that was factorized to estimate the embeddings. It turns out that alpha controls the distance between the words in the embedding transformation process. Next the paper discusses how to choose/estimate alpha to get better quality embeddings. Results are shown on several word similarity tasks.


Comments:
=======
The paper offers fresh insights into the well studied problem of learning word embeddings. The impact of the alpha parameter is definitely interesting w.r.t the quality of embeddings learned. That said, the paper does have a few problems. First, though the paper is well motivated and puts itself nicely in context of previous work, it needs a copy-editor as there are many language/grammar issues some of which I highlight below.


Second, and the main problem with the paper, is that the properties of the alpha parameter are intriguing but the experimental evaluation is underwhelming. The paper also needs to show the impact of the alpha parameter on the quality of embeddings learned for some downstream task e.g. NER, POS Tagging. Just showing results on word similarity tasks and computing correlations is not very insightful or useful.


Grammar issues (subset):

Page 1: "Word embedding is a very important task"

Page 1 : "...which value should alpha be?"

Page 1: "..has an important influence to the..."

Page 6: "The first is to verify...."


**Experience Assessment:**

I have published one or two papers in this area.

**Review Assessment: Checking Correctness Of Derivations And Theory:**

I assessed the sensibility of the derivations and theory.

**Review Assessment: Checking Correctness Of Experiments:**

I carefully checked the experiments.

**Review Assessment: Thoroughness In Paper Reading:**

I read the paper thoroughly.

---

> ### Author Response · Authors · 2019-11-15
> **Thanks a lot for your review.**
>
> Firstly, thanks a lot for your comments! Here are the key points:
>
> 1. About the grammar issues.
> We correct these grammar typos according to your comment.
>
> 2. About other downstream tasks.
> Firstly, we agree with you that it might be better to conduct more experiments on the downstream task to further show the impact of the alpha parameter. However, we have to say that the influence of alpha has been empirically observed by previous studies (e.g. Bullinaria and Levy (2012), Levy et al. (TACL'15), Artetxe et al. (CoNLL'18)), and our purpose is not to empirically re-examine this phenomenon. Differently, we focus on the theoretical understanding of the inner reason why word embedding exhibits these two well-known properties, namely the word similarity and the analogy structure. Particularly, we theoretically explain how ALPHA influences the word embedding. The purpose of the conducted experiments is to verify these theoretical analysis.

---

### Official Review · AnonReviewer2 · 2019-10-27
**Official Blind Review #2**

**Rating:** 6

**Review:**

In this paper, the authors study the word embedding, with a particular emphasize on the word2vec or similar strategies. To this end, the authors consider the matrix factorization framework, previously introduced in the literature, and also study the influence of an hyperparameter denoted by alpha. Roughly speaking, there are two major parts in the paper. On one hand, it explains the reasons why the word embedding schemas provide nice properties, by defining the embedding as a low rank transformation mechanism. On the other hand, they propose to choose optimally the hyperparameter alpha in order to ameliorate word embedding by better preserving the distance structure. Conducted experiments are convincing.

The paper is well written, and derivations seem correct.

We think that the major issue in this work is that it does not provide significant contributions with respect to the state of the art. It seems that the contributions are rather incremental compared to related works, such as Levy et al. 2015 and Yin & Shen 2018. In the latter, it is proven that most existing word embedding schemas can be formulated as low rank matrix approximations, either explicitly or implicitly. The submitted paper does not provide new significant results.

Moreover, the authors have failed to provide connections to other related works, or even cite them, including papers that consider word embedding as asymmetric low-rank projections. See for example:
Fei Tian, Bin Gao, Enhong Chen, Tie-Yan Liu
Learning Better Word Embedding by Asymmetric Low-Rank Projection of Knowledge Graph
J. Comput. Sci. Technol. (2016) 31: 624.
https://doi.org/10.1007/s11390-016-1651-5
Also available on ArXiv: https://arxiv.org/abs/1505.04891


------
Reply to Rebuttal

We thank the authors for modifying the paper and the reply to out comments and suggestions. However, we still think that the paper is of low quality, due to straightforward extension to the paper of Levy et al from 2015.

The authors have added a small section on related works, as recommended. However, they have removed the "Conclusion" section. The paper no longer has a conclusion and potential work that ends the paper.

**Experience Assessment:**

I have read many papers in this area.

**Review Assessment: Checking Correctness Of Derivations And Theory:**

I did not assess the derivations or theory.

**Review Assessment: Checking Correctness Of Experiments:**

I assessed the sensibility of the experiments.

**Review Assessment: Thoroughness In Paper Reading:**

I made a quick assessment of this paper.

---

> ### Author Response · Authors · 2019-11-15
> **Thanks a lot for your review!**
>
> 1. About the contributions and the connection with previous works
> While we adopted the matrix factorization framework that was once proposed (Levy, et al. (NeurIPS'14)) to discuss word embedding, our focus is different from the previous studies. As mentioned in your review, we provide the theoretical explanation why the word embedding exhibits nice properties. Besides, while previous studies (e.g. Bullinaria et al. (2012), Levy et al. (2015), Artetxe et al. (CoNLL'18)) empirically observed  the parameter $\alpha$ influences the quality of word embedding, they do not provide theoretical explanation or the method about how to set this parameter. On the contrary, we provide a theoretical explanation for this behavior, and derive a method to automatically find its optimal value. In detail, the relation between this paper and previous works are summarized as follows:
> (1) The relation between this paper and Levy et al (NeurIPS'14) is that Levy et al (NeurIPS'14) proved that SGNS is implicitly factorizing the (shifted) PMI matrix.  In this paper, we adopt this matrix factorization framework to discuss the word embedding. With this assumption, we provide theoretical explanation about the word similarity and analogy structures existing in the word embeddings (e.g. SGNS).
> (2) The relation between this paper and Bullinaria et al. (2012), Levy et al. (2015), Artetxe et al. (CoNLL'18) and other existing words that discussed the parameter $\alpha$ is that, these existing methods empirically found that the parameter $\alpha$ can influence the quality of the learnt word embeddings. But these methods did not give the theoretical explanation to explain why $\alpha$ has such influence and they did not give out a clear method to guide us to find the optimal $\alpha$. In this paper, we theoretically explain how $\alpha$ influences the word embedding, and provide the method as a guide to find the optimal $\alpha$.
> (3) The relationship between this paper and Yin & Shen (2018) is that, Yin & Shen and this paper both discuss the word embedding under the matrix factorization framework. Yin & Shen mainly focuses on the dimensionality of the word embedding such that how to choose the dimension of the learnt word embedding, while this paper focuses on the theoretical explanation about the inner mechanism of the word similarity and analogy structure existed in the word embedding. Besides, section 5.1 in Yin & Shen's paper mentioned that $\alpha$ and they suggest that $\alpha$ should be larger because they states that "as $\alpha$ becomes larger, the embedding algorithm becomes less sensitive to over-fitting caused by the selection of an excessively large dimensionality k". However, in the real case, the very large $\alpha$ does not produce the embedding with the best quality. This point is verified in our experiments and the experiments in Bullinaria et al. (2012), Levy et al. (2015), Artetxe et al. (CoNLL'18), etc,.
>
> 2. About the mentioned related work.
> We are sorry we once missed it. We add it into the revised version.

---

### Decision · Program_Chairs · 2019-12-19

**Decision:**

Reject

**Comment:**

The paper studies word embeddings using the matrix factorization framework introduced by Levy et al 2015. The authors provide a theoretical explanation for how the hyperparameter alpha controls the distance between words in the embedding and a method to estimate the optimal alpha.  The authors also provide experiments showing the alpha found using their method is close to the alpha that gives the highest performance on the word-similarity task on several datasets.

The paper received 2 weak rejects and 1 weak accept.  The reviews were unchanged after the rebuttal, with even the review for weak accept (R2) indicating that they felt the submission to be of low quality.  Initially, reviewers commented that while the work seemed solid and provided insights into the problem of learning word embeddings, the paper needed to improve their positioning with respect to prior work on word embeddings and add missing citations.  In the revision, the authors improved the related work, but removed the conclusion.

The current version of the paper is still low quality and has the following issues
1. The paper exposition still needs improvement and it would benefit from another review pass
Following R3's suggestions, the authors have made various improvements to the paper, including modifying the terminology and contextualizing the work.  However, as R3 suggests, the paper still needs more rewriting to clearly articulate the contribution and how it relates to prior work throughout the paper.  In addition, the conclusion was removed and the paper still needs an editing pass as there are still many language/grammar issues.

Page 5: "inherites" -> "inherits"
Page 5: "top knn" -> "top k"

2. More experimental evaluation is needed.
For instance, R1 suggested that the authors perform additional experiments on other tasks (e.g. NER, POS Tagging).  The authors indicated that this was not a focus of their work as other works have already looked at the impact of alpha on other task.  While prior works has looked at the correlation of alpha vs performance on the task, they have not looked at whether alpha estimated the method proposed by the author will give good performance on these tasks as well.  Including such analysis will make this a stronger paper.

Overall, there are some promising elements in the paper but the quality of the paper needs to be improved.  The authors are encouraged to improve the paper by adding more experimental evaluation on other tasks, improving the writing, as well as incorporating other reviewer comments and resubmit to an appropriate venue.